# The Role of Fructose in Non-Alcoholic Steatohepatitis: Old Relationship and New Insights

**DOI:** 10.3390/nu13041314

**Published:** 2021-04-16

**Authors:** Alessandro Federico, Valerio Rosato, Mario Masarone, Pietro Torre, Marcello Dallio, Mario Romeo, Marcello Persico

**Affiliations:** 1Department of Precision Medicine, University of Campania Luigi Vanvitelli, 80138 Naples, Italy; marcello.dallio@unicampania.it (M.D.); mario.romeo@unicampania.it (M.R.); 2Internal Medicine and Hepatology Division, Department of Medicine, Surgery and Odontostomatology, “Scuola Medica Salernitana”, University of Salerno, 84084 Salerno, Italy; valeriorosato@gmail.com (V.R.); mmasarone@unisa.it (M.M.); pietro.torre@gmail.com (P.T.); mpersico@unisa.it (M.P.); 3Liver Unit, Ospedale Evangelico Betania, 80147 Naples, Italy

**Keywords:** non-alcoholic steatohepatitis, fructose, metabolism, nutrients

## Abstract

Non-alcoholic fatty liver disease (NAFLD) represents the result of hepatic fat overload not due to alcohol consumption and potentially evolving to advanced fibrosis, cirrhosis, and hepatocellular carcinoma. Fructose is a naturally occurring simple sugar widely used in food industry linked to glucose to form sucrose, largely contained in hypercaloric food and beverages. An increasing amount of evidence in scientific literature highlighted a detrimental effect of dietary fructose consumption on metabolic disorders such as insulin resistance, obesity, hepatic steatosis, and NAFLD-related fibrosis as well. An excessive fructose consumption has been associated with NAFLD development and progression to more clinically severe phenotypes by exerting various toxic effects, including increased fatty acid production, oxidative stress, and worsening insulin resistance. Furthermore, some studies in this context demonstrated even a crucial role in liver cancer progression. Despite this compelling evidence, the molecular mechanisms by which fructose elicits those effects on liver metabolism remain unclear. Emerging data suggest that dietary fructose may directly alter the expression of genes involved in lipid metabolism, including those that increase hepatic fat accumulation or reduce hepatic fat removal. This review aimed to summarize the current understanding of fructose metabolism on NAFLD pathogenesis and progression.

## 1. Introduction

Τροφὴ οὐ τροφή, ἢν μὴ δύνηται: μὴ τροφὴ τροφή, ἢν οἷόν τε ᾖ τρέφεσθαι. οὔνομα τροφή, ἔργον δὲ οὐχί: ἔργον τροφή, οὔνομα δὲ οὐχί
*A nutriment is not a nutriment, if it does not have its power. Is not a nutriment a nutriment, if it is impossible to be nourish by it. Nutriment in name, not in deed; nutriment in deed, not in name.*
Ippocrates, *De Alimento* (ΠEPI TPOΦHΣ)

Glucose represents the basis of the majority of metabolic processes in animals, while the sucrose, the disaccharide composed by equal proportion of glucose and fructose, is the main sugar present in plants and, therefore, it constitutes the predominant form of sugar used by herbivores and omnivores for their energy and biosynthetic needs [1].

Natural food is characterized by a small amount of fructose and, due to its slow absorption, after the consumption of fruits or vegetables, its serum increase is negligible [2].

Numerous manufactured foods largely diffused in Western dietary regimens are characterized by sucrose or high-fructose corn syrup (HFCS) (composed by 45% glucose and 55% fructose) supplementation. The addition of HFSC in sugar-sweetened beverages (SSBs) is widely used by food industries due to its low cost and easy chemical manageability compared with sucrose [3]. SSBs are estimated to be a major source of added sugar in daily diet, contributing to 7% of the total daily calories and nearly 50% of sugars [4].

In a recent prospective study, an increased SSBs consumption over 6 years of follow-up was associated with huge increase of visceral adipose tissue volume, one of the most important cardio-metabolic risk factors [5]. However, this effect was not only limited to adipose tissue due to the evidence of liver fat accumulation derived from the consumption of 1 L of SSBs over 6 months in overweight subjects [6]. It is known that a high-glucose diet, determining the increase of glycemia and insulinemia, can mediate the adverse events associated with type 2 diabetes (T2DM) and cardiovascular diseases (CVD) [7].

However, among the components of SSBs and manufactured food with added sugar, fructose appears to be particularly dangerous due to its deleterious metabolic effects exerted via different mechanisms than those of glucose. In fact, on the basis of the results of studies on fructose overfeed animal models and short-term dietary studies in humans, it was demonstrated that a high fructose consumption (25% of energy requirement), but not glucose, may increase multiple features of metabolic syndrome (MetS), such as visceral adiposity, postprandial hypertriglyceridemia, and insulin resistance, by acting on hepatic de novo lipogenesis [8]. In a meta-analysis on 15 studies, fructose consumption was positively associated with increased fasting blood sugar, elevated triglycerides, and systolic blood pressure [9]. A controversial element in contrast with this evidence is represented by the amount of fructose evaluated in these studies that exceeds what is commonly consumed in ad libitum diets, on average 9% of total energy intake up to 15% in the 95th percentile of the US population [10]. However, in a short-term interventional study, the metabolic effects of SSBs contained in high-fructose corn syrup were determined on 85 volunteers showing a dose-dependent increase of lipid/lipoproteins (risk factors for CVD: LDL cholesterol, non-HDL cholesterol, ApoB, ApoCIII) and uric acid, which occurred within 2 weeks of administration, and even fructose was in the “range” of normal consumption [11].

Unfortunately, long-time interventional study assessing the effects of added sugar on cardio-metabolic risk, probably due to complexity, cost, and potential ethical issues, are still lacking.

Moreover, some results actually remain controversial, for instance, low doses of natural fructose (<20 g/d) showed an improvement or no effects on MetS parameters [12]. Nevertheless, much scientific evidence has established that an excess in fructose consumption is a causative factor for insulin resistance and fatty liver onset, thus conducing the increase of nonalcoholic fatty liver disease (NAFLD) incidence [13,14].

In the last decades, at the same time as the “Western lifestyle” has been adopted in many countries, NAFLD has become the most frequent liver disease worldwide [15]. It encompasses a large spectrum of liver diseases that range from simple fatty liver to non-alcoholic steatohepatitis (NASH) to cirrhosis and its complications, such as hepatocellular carcinoma (HCC) [16]. Until now, cause, pathophysiology, and pathogenesis of NAFLD have remained partially unclear, even if strong association with features of MetS or T2DM is well established [17]. As proof of this close correlation, it was shown that the coexistence of MetS and T2DM increases the risk of NASH and, consequently, worsens liver fibrosis [18]. In this picture, the fructose may play the role of a common etiological factor for all the above-mentioned conditions.

Moreover, animal studies showed that a fructose-enriched diet results in an increased hepatic triglycerides content, reproducing the histological features of NASH in an undeniable way in comparison to glucose- or fat-enriched diets [18,19].

As a matter of fact, it was found that subjects with NAFLD have, in their dietary history, about a two-fold higher intake of SSBs compared to the general population [20].

Furthermore, fructose is also associated with the severity of NAFLD. In a study on 427 biopsy-proven NAFLD, fructose consumption was shown to be associated with higher fibrosis stage. Similarly, in 271 NAFLD pediatric patients, a more severe NASH phenotype was found in those with higher fructose consumption [21,22]. While the majority of NAFLD patients are overweight or obese, this disease could develop also in normal weight patients, who represent a clinical subtype named “lean NAFLD” [23].

Lean NAFLD patients could also present a severe histological hepatic disease, with higher mortality and morbidity compared to overweight or obese [24]. In a prospective observational study, non-diabetics or overweight NAFLD patients self-reported excessive SSBs consumption (more than 50 g/day). Moreover, among all the risk factors taken into consideration (including dietary composition and physical activity), SSBs intake was the only highlighted independent NAFLD associated variable [25]. Similarly, in another study in which NAFLD patients not presenting metabolic syndrome risk factors were analyzed, SSBs consumption three times higher than in healthy controls was reported, highlighting that SSBs consumption was significantly associated with higher risk to develop fatty liver disease (OR = 2) [26]. In addition to the hepatic metabolism derangements, it was also described that fructose could induce habituation and possibly addiction, similarly to the effects of ethanol on humans [6].

On the basis of this evidence, a “fructose hypothesis” was formulated in order to explain the potential mechanisms for NAFLD development and progression.

The aim of this review is represented by the analysis of the actual knowledge on the relationship between fructose metabolism and NAFLD pathogenesis, giving the new insights for the management of the disease.

## 2. Fructose Metabolism and NAFLD Development

### 2.1. Difference between Glucose and Fructose Metabolism

Despite having exactly the same molecular formula (C6H12O6), glucose and fructose require distinct transporters for their intestinal absorption and have different metabolic pathways (Figure 1).

Glucose is absorbed by intestinal epithelial and hepatic cells via the glucose transporter type 2 (GLUT2). In the cytosol, glucose generates glucose 6-phosphate (G6P) through glucokinase (GK) activity, and G6P is successively converted to glycogen for energy storage or metabolized via glycolysis for adenosine triphosphate (ATP) and pyruvate production [27]. The pentose phosphate pathway (PPP) uses G6P as a precursor to generate nicotinamide adenine dinucleotide phosphate (NADPH) and ribose-5-phosphate, which are crucial for nucleotide synthesis [27]. The excess of glucose produces acetyl-CoA overload that, through the citric acid cycle, acts as a precursor for fatty acids synthesis. A limiting step of glycolysis is mediated by phosphofructokinase (PFK) that is inhibited by ATP and citrate in case of low energy expenditure, thus limiting the glucose uptake and the production of substrates for de novo lipogenesis [1].

However, the greater part of the ingested glucose bypasses the liver reaching the systemic circulation and stimulating the release of insulin that, in turn, activates via insulin receptor substrates (IRS 1/2) the glycogen synthesis and regulates hepatic glucose production [17]. Fructose requires very distinct transporters for intestinal absorption (GLUT2 and GLUT5). GLUT2, located in the basolateral side of the intestinal epithelial cells, shows low affinity for fructose. In fact, a study on GLUT2 whole-body knockout mice showed only mildly decreased fructose absorption [28]. GLUT5, located in the apical side of intestinal cells, is highly expressed in the small intestine and demonstrated high affinity for fructose, diffusing it rapidly into intestinal capillaries for its transport to the liver via the portal vein [29]. GLUT5 whole-body knockout mice, despite surviving without any defects under typical chow diets, showed lethal phenotype upon fructose feeding, developing a severe fructose intolerance characterized by large intestine distension and fluid retention [30].

Recent data showed that high-fructose diets indirectly upregulate the intestinal GLUT5, inducing intestinal thioredoxin-interacting protein (TXNIP) and resulting in increased fructose intestinal absorption [31]. The TXNIP gene expression is regulated by fructose through the activation of carbohydrate-responsive element binding protein (ChREBP), a transcriptional factor also involved in glycolytic, pentose phosphate, and de novo lipogenic pathways [32].

As proof of the diet-induced increase of intestinal absorption, it was shown that children with NAFLD demonstrated higher levels of fructose absorption than their lean counterparts [33]. On the other hand, GLUT2 and GLUT8 are the major contributors to hepatic fructose uptake [34]. Unlike what happens with glucose, after fructose consumption, its concentration in peripheral plasma undergoes a rapid clearance due to a more effective hepatic extraction, whereas with the liver extract with 15–30% of an oral glucose load, the hepatic fructose extraction may reach up to 70% [35]. In fact, unlike glucose, fructose bypasses the critical regulation step mediated by PFK and, due to the lack of any regulation by circulating hormones or negative feedback, it is rapidly phosphorylated to fructose-1-phosphate (F1P) by ketohexokinase (KHK; fructokinase) [36]. In this way, the hepatic fructolysis is unrestricted and under high fructose loads may lead to high increase in substrates for some metabolic pathways, including glycolysis, glycogenesis, gluconeogenesis, lipogenesis, and oxidative phosphorylation.

KHK has two splice variants, KHK-A and KHK-C, of which the former is ubiquitously expressed at low levels but is much less effective for fructose phosphorylation than KHK-C, which shows ten times greater affinity for fructose, making it of primary importance in fructolysis [37].

The high activity and the refractoriness to cellular energy status of KHK maintain the plasma gradient of fructose constantly low, allowing for continuous fructose uptake in hepatocytes by GLUT2 [38]. Furthermore, F1P exerts a strong positive regulatory control on GK, promoting its release from inhibitory GK regulatory protein, inducing then the glucose uptake and leading to rapid glycogen or lipid accumulation [34]. In this sense, the fructose-derived F1P and its regulation on GK acts as an evolved hepatic signaling mechanism for the amount of sugar consumption, enhancing hepatic glycogen and lipid storage in case of fructose-containing sugar consumption.

Interestingly, pathological levels of serum glucose may induce the release of GK similarly to G1P, indicating that, in case of diabetes or increased sugar consumption, it may exert similar effects on the liver [39]. Aldolase B hydrolyzes F1P in dihydroxyacetone (DHAP) and glyceraldehyde, which enter in the glycolytic/gluconegenic metabolites pools as glyceraldehyde-3-phosphate [40].

DHAP can also be reduced to glycerol-3-phosphate, resulting in useful de novo lipogenesis for triglycerides and lipoproteins synthesis [1]. Aldolase B is the principal enzyme responsible for F1P cleavage. Indeed, it was shown that aldolase B knockout mice, similarly to GLUT5 knockout mice, demonstrated high death rates in cases of high-fructose diets, developing severe hepatic fat accumulation and fibrosis [41].

Because of these metabolic differences, it is supposed that fructose may induce hepatic lipogenesis in a more powerful manner in comparison to glucose. Indeed, in a high-fat diet mice study, it was shown that 30% fructose-sweetened water led to more severe NAFLD manifestation compared to isocaloric diet with 30% glucose-sweetened water [11].

Moreover, fructose exerts greater hepatotoxicity compared to glucose, an effect mainly based on insulin resistance, oxidative stress, and the GUT microbiota dysbiosis.

### 2.2. Insulin Resistance

Fructose is responsible for the activation of several pathways involved in lipogenesis, gluconeogenesis, and glycolysis, irrespective of negative feedback regulations link to insulin signaling (Figure 2) [42].

Moreover, it seems to be more potent and faster than glucose in inducing the biogenesis of advanced glycation end-products, such as hemoglobin A1c, indicating its involvement in diabetes complication onset [43,44].

As a matter of fact, it was demonstrated that, even if fructose by itself does not induce pancreatic insulin exocytosis because of the lack of GLUT5 on the surface of pancreatic ß-cells, the constant exposure to high fructose levels results in hyper-reactivity of pancreatic ß-cells in response to glucose [45]. In fact, in a rodent high-fructose dietary model, a more pronounced hyperinsulinemia compared with a high-dextrose dietary regimen was reported and, similarly, it was demonstrated that hypercaloric fructose feeding increases circulating insulin levels in humans [46,47].

Moreover, a meta-analysis analyzing the effects of fructose on insulin sensitivity collected several studies based on short-term fructose consumption on nondiabetic subjects and demonstrated the development of hepatic insulin resistance, in cases of both eucaloric and hypercaloric addition in the dietary regimen [47]. However, the same studies did not show a worsening in peripheral insulin sensitivity, measured with fasting plasma insulin concentration or homeostasis model assessment of insulin resistance (HOMA-IR) [47]. Nevertheless, chronic hyperinsulinemia in response to fructose-induced hyperglycemia could itself induce peripheral insulin resistance, but this hypothesis needs further confirmation by experimental studies [48]. As far as lipogenesis is concerned, an excessive fructose consumption induces hepatic lipid accumulation, increasing DNL, or esterification of preformed fatty acids, and decreasing of VLDL secretion, or hepatic fatty acids oxidation. It was demonstrated that the liver fat accumulation is characterized by several features of insulin resistance in both normal weight and overweight subjects [18].

The activation of this lipogenic metabolic programing was observed immediately after a single administration of fructose in human studies, in which an increase in intrahepatocellular lipids (evaluated by magnetic resonance spectroscopy) or an increase in DNL (evaluated by dosing the percentage of palmitate, significantly correlated with DNL activity) were reported [49,50]. Fructose contributes to DNL, not only increasing glycerol 3-phosphate, used as a metabolite for hepatic triglycerides production, but also activating, especially in case of chronic consumption, key transcriptional factors for DNL, such as sterol regulatory element-binding protein 1c (SREBP1c) and carbohydrate-responsive element-binding protein (ChREBP) [51]. SREBP1c and ChREBP are potent inducers of lipogenesis by mean of triggering the activation of genes such as fatty acid synthase (FAS) and acetyl coA carboxylase (ACC) [38]. In this way, fructose acts as the substrate and the activator of DNL, representing a potent lipogenic carbohydrate that contributes to the development of liver steatosis. SREBP1c is mainly activated by insulin and hyperinsulinemic state, secondary to chronic fructose ingestion, even if in an insulin receptor-knockout mouse model of high fructose dietary regimen with an independent activation of SREBP1c was demonstrated as well [18,52]. ChREBP, in addition to the induction of several enzymes required for DNL, suppresses fatty acid oxidation through the downregulation of carnitine palmitoyltransferase 1a, limiting, then, the translocation of fatty acids into the mitochondria [53]. Nowadays, the mechanism by which sugar metabolites activate ChREBP remains still not fully understood, even if, in murine studies, it was shown that high-fructose diet induces higher hepatic expression and activity of ChREBP compared to isocaloric high-glucose diet [54,55].

In ChREBP knockdown fructose-fed rats, a low level of circulating triglycerides was demonstrated, confirming the role of ChREBP in fructose-mediated dyslipidemia but not on steatosis development, highlighting that fat accretion in lipid droplets and very low-density lipoprotein (VLDL) secretion are distinct processes [56]. However, ChREBP has a role both in glucose and lipid homeostasis by regulating metabolic genes expression of key enzymes, such as pyruvate kinase and GLUT2 involved in glycolysis, and transketolase, which is involved in the pentose phosphate pathway [57]. Moreover, fructose particularly upregulates, via ChREBP, the gene expression and the activity of glucose-6-phosphatase (G6Pase), increasing the endogenous glucose production [54].

Furthermore, as evidence of the independence of fructose from suppressive effects of insulin, the ChREBP-mediated induction of G6Pase also occurs in absence of the transcription factor forkhead box O1a (FoxO1), which transactivates gluconeogenic enzymes expression and is inhibited by insulin [54]. Therefore, the indirect gluconeogenic and lipogenic action of fructose is not affected by the inhibitory effects of insulin, impairing the hepatic insulin resistance. ChREBP also regulates the expression of fibroblast growth factor (FGF)-21, a secretory hormone induced by fasting that modulates simple sugars intake and sweet taste preference, producing an endocrine satiety signal and reducing sugars intake [58,59].

Some studies on murine models demonstrated that high-fructose diet is also able to induce a reduction in the phosphorylation of insulin receptors substrate (IRS)-1 and the expression of IRS2 and FoxO1, resulting in the inhibition of insulin signaling and the increase of plasma glucose excursions during glucose and pyruvate tolerance tests [60,61].

However, despite the amount of evidence regarding direct and indirect effects of fructose on insulin signaling, strong scientific evidence derived from well-constructed clinical studies is still required.

### 2.3. Oxidative Stress

Beside the induction of lipogenesis, fructose metabolism exerts other specific hepatotoxic effects by inducing an increase in oxidative stress (Figure 3) [42].

Whereas fructose by itself or through DNL could promote oxidative stress, the development of the inflammatory processes that leads to the progression from simple steatosis to NASH is mainly mediated by mitochondrial dysfunction and endoplasmic reticulum (ER) stress [62].

Fructose could directly generate reactive oxygen species (ROS) and promote hepatocellular damage by mean of protein fructosylation, which is seven times faster than glycation by glucose [62].

In a mouse model, a prolonged high-fructose diet, compared to a high-glucose one, demonstrated the ability to induce higher hepatic accumulation of carboxymethylisine, a glycation product that can induce lipogenesis via SREBP1 activation [63]. Similarly, hepatic metabolism of fructose was demonstrated related to the production of methylglyoxal, another potent glycating agent that leads to altered insulin signaling and cellular stress [64,65]. The rapid phosphorylation of fructose in F1P mediated by KHK is ATP-dependent, thus the ingestion of fructose leads to ATP depletion and the subsequent induction of AMP deaminase to stimulate nucleotide degradation in order to restore ATP levels. These events result in the accumulation of uric acids produced by xanthine oxidase activity [66]. Indeed, it was shown that high-fructose intake increases the risk of hyperuricemia and gout, which are closely associated with NAFLD [67].

The consequent high amount of uric acid in the hepatocytes causes mitochondrial oxidative stress with ROS production and increased citrate release, which leads to de novo synthesis of triglycerides from acetyl-CoA through ATP citrate lyase and fatty acid synthase induction [68]. It was demonstrated that uric acid causes ER stress, determining NAFLD development after the disruption of ER homeostasis, the activation of unfolded protein response (UPR) and other pro-inflammatory pathways [69]. In fact, uric acid stimulates, by generating ROS, NADPH oxidase activity which, in turn, is involved in lipid metabolism because of the induction of the transcriptional factors ChREBP and SREBP-1 that activate, in turn, the lipogenic genes ACC and FAS [70].

Uric acid induces NADPH oxidase, increasing mitochondrial ROS production and generating a worsening, vicious circle [70]. In fact, the uric acid released by hepatocytes can be absorbed by adipocytes, in which it activates NAPDH oxidase, generating ROS and oxidative stress and contributing to DNL, insulin resistance, and NAFLD [71,72]. Fructose could directly induce the fructosylation of ER membrane, or, alternatively, the enhanced synthesis of triglycerides driven by fructose could overload the ER membrane, leading to ER stress and UPR [62,73]. ER stress contributes to the progression of hepatic steatosis and insulin resistance by activating DNL (via the protein kinase RNA-like ER kinase (PERK)/activating transcription factor 4 (ATF4)/eukaryotic translation initiation factor 2alfa (eIF2alfa) pathway) and limiting the secretion of VLDL (via inositol-requiring signaling protein 1 (IRE1)) [69,74]. Finally, ER stress promotes the initiation of inflammatory and apoptotic pathways through c-Jun N-terminal kinase (JNK), nuclear factor kB (NFkB), and CCAAT/enhancer-binding homologous protein (CHOP), which play an important role in NAFLD progression [69,75].

### 2.4. Inflammation and NASH

The lipogenic mechanism and the oxidative stress induced by fructose consumption contribute to hepatic inflammation and, consequently, progression to NASH (Figure 3). As proof of the specific fructose role in determining liver inflammation, it was demonstrated that fructokinase knockout mice (KHK-A and KHK-C) under high-fat high-sucrose diets did not develop steatohepatitis compared to wild-type mice, despite both groups being characterized by obesity and mild hepatic steatosis onset [76]. Kupffer cells critically contribute to NAFLD progression, amplifying ROS-induced inflammation and toll-like receptor (TLR), in particular TLR4, activation [77].

TLR4 may be activated by altered lipid homeostasis and fatty acids such as palmitate, production of which may be promoted by fructose [77]. The activation of TLR4 stimulates nitric oxide synthase and NF-kB, inducing the production of pro-inflammatory cytokines such as tumor necrosis factor alfa (TNF-α) [62].

The activation of TLR4 may also activate NOD-, LRR-, and pyridine domain-containing protein 3 (NLRP3) inflammasome, an intracellular multiprotein complex involved in the production of interleukin (IL) 1-beta and interleukin 18 [78]. NLRP3 plays a crucial role in the progression of liver fat overload in NASH, activating immune system modulators responsible for fibrosis stimulation and inflammation [78,79]. NLP3 knockout mice, in a choline-deficient amino acid-defined (CDAA) diet induced with NAFLD, were protected against liver injury, hepatic inflammation, and liver fibrosis, while, after 4 weeks of CDAA diet, they showed severe liver inflammation with huge infiltration of activated macrophages and early signs of liver fibrosis [18]. The accumulation of uric acids induced by fructose generates ROS, resulting in the release of pro-inflammatory cytokines [80]. Furthermore, xanthine oxidase, the last enzyme involved in uric acid synthesis induced by the excess of substrates supplied by fructose, may act as an electron donor for oxygen, thus generating ROS and fueling liver oxidative stress [18,81]. Uric acid may also directly activate the inflammasome NLRP3, exerting the aforementioned detrimental effects [78]. The increasing amount of cytokines, such as TNF-α, IL-1beta, and IL-18, produced in hepatocytes and adipocytes in response to the excess of fructose consumption, may contribute to the activation of the hypothalamus–pituitary–adrenal (HPA) axis, resulting in the release of immunosuppressors as glucorticoids [82]. The increased activity of cortisol may induce insulin resistance, reducing the lipogenesis in subcutaneous adipocyte tissue and leading to visceral and hepatic fat deposition [82].

The fructose induced inflammation in visceral adipose tissue increases the activity of 11-beta-hydroxysteroid dehydrogenase type 1 (11beta-HSD-1), an isoenzyme involved in the conversion of cortisol from cortisone, resulting in a more severe expression of metabolic syndrome and contributing to the progression of NASH [83].

Furthermore, GLUT 5 transporter present in the blood–brain barrier allows fructose to enter in the central nervous system, triggering the phosphorylation of the energy sensor of AMP-activated protein kinase in neurons of the paraventricular nucleus of the hypothalamus; the fructose may induce the release of corticotropin-releasing hormone (CRH) and, subsequently, of adrenocorticotropic hormone (ACTH), stimulating the corticosteroids production [84]. In this way, the fructose centrally induces glucorticoid stress response, triggering, in turn, liver de novo gluconeogenesis and defining the mechanism of “sugar making sugar” [84]. Fructose also modulates liver inflammation, influencing the gut microbiota homeostasis, as discussed in detail below.

## 3. Intestinal Absorption of Fructose and Microbial Fructose Metabolism in NAFLD

Regarding fructose metabolism triggering NAFLD, it is well established that intestinal absorption may regulate the load of fructose reaching the liver and microbial dysbiosis, increasing endotoxin levels, intestinal permeability, and progression of liver damage [42,85].

A study used stable isotopes of ^13^C-fructose or glucose to trace, in a mouse model, fructose and glucose intake and metabolism. It showed that, in small intestine, low doses of dietary fructose intake (0.25–0.5 g/kg) were almost completely cleared and transformed in glucose. Instead, high doses (>1 g/kg) overwhelmed the intestinal fructose absorption and clearance, resulting in fructose reaching both the liver and the colonic microbiota [86]. Importantly, the clearance of fructose exerted by small intestine was augmented by a priori exposure to fructose or food consumption [86].

In a recent study on intestinal-specific KHK-C knockout mice, generated in order to abolish intestinal fructose catabolism, higher hepatic lipogenesis with worse fatty liver and hyperlipidemia compared to mice with KHK-C overexpression was demonstrated [87]. Furthermore, the effect of high-dose fructose bolus (2 g/kg) showed higher fructose spillover into portal circulation and induction of hepatic lipogenic genes compared to slit doses (0.5 g/kg × 4) [87]. Therefore, it can be assumed that the small intestine acts as a shield for the liver, protecting it from an excessive fructose load. However, a previous study on 36 lean, healthy men found that the high frequency of hypercaloric high-fat and high-sugar meals led to higher increase of intrahepatic triglycerides content and lower decrease of insulin sensitivity rather than a bigger meal size [88].

Intestinal dysbiosis may contribute to the pathogenesis of NAFLD, and it was identified, through meta-genome sequencing and metabolomics technologies, that differential gut microbiota subpopulations and their metabolic products may have greater influence on the development and the progression of liver disease [89,90]. A great deal of evidence in literature highlighted that fructose consumption may affect the gut permeability and the microbiome, increasing endotoxin translocation [91,92]. Moreover, in studies on rodents under high-fructose diet, a reduction of tight junction with a subsequent increased delivery of bacterial endotoxins, such as lipopolysaccharide (LPS), into the portal circulation was shown [93]. In pediatric cohorts of NAFLD patients, acute and chronic exposure to high fructose dietary regimens increased the endotoxin levels, markers of insulin resistance, and several inflammatory cytokines compared to obese controls [94]. Furthermore, in non-human primates, it was shown that fructose, even in the absence of weight gain, increased microbial translocation, endotoxin levels in plasma, and liver damage [95]. LPS is the most common pathogen-associated molecular pattern (PAMP), and it may induce hepatic inflammation binding and activation of TLR4 [96]. F11r knockout mice encoding junctional adhesion molecule under high-fructose diets showed upregulation of TLRs and inflammatory contentment, developing more severe steatohepatitis compared to normal diet mice control [97].

Thus, the rise in LPS serum level induced by fructose and the subsequent liver TLR4 expression and inflammation have key roles in the pathogenesis of NAFLD. It was shown that mice with fructose and/or fat induced steatohepatitis were characterized by elevated endotoxin and TLR4 levels, which were associated with Kupffer cells activation and macrophages recruitment [98]. Furthermore, in the same study, the alterations of gut permeability were characterized by a reduction level of the tight junction protein occludin and of the zonula occludens 1 in the duodenum [98].

The small intestinal fructose absorption is limited, thus, after a high dose of fructose intake, a huge dose of this carbohydrate may reach the large intestine, inducing bacterial fermentation [99,100]. The fructose is phosphorylated by gut microbiota hexokinase in F6P for subsequent glycolysis [42]. It may be hypothesized that some intestinal microbiome subpopulations exert high fructose catalytic ability, outcompeting with others or forming cooperative relationships with those having no fructose metabolic capabilities, promoting intestinal dysbiosis [101]. Primarily, six different bacterial phyla (*Firmicutes, Bacteroides, Proteobacteria, Actinobacteria, Fusobacteria, and Verrucomicrobia*) colonize the healthy gut, but *Firmicutes* (gram-postive), *Bacteroides* (gram-negative), and *Actinobacteria* (gram-positive) represent about 90% of total gut adult microbiota [102]. Several studies on mice described the effect of high fructose diet on *Firmicutes/Bacteroides* ratio in NAFLD context, showing a decrease in relative abundance of *Bacteroides* [91,96]. *Bacteroides* usually dominates the intestinal tracts in healthy individuals, and their reduction is associated with a detrimental effect on NAFLD development [96]. Furthermore, different studies on diet-induced obesity in mice suggested that an abundance of *Firmicutes* promotes body fat accumulation, increasing the capacity to harvest energy from the diet [103,104]. In rats with fructose-induced metabolic syndrome, *Firmicutes* were much more abundant than *Bacteroides* and *Proteobacteria* [105].

However, there is still controversy over which species is more detrimental. In fact, a changed *Firmicutes/Bacteroides* ratio in favor of *Bacteroides* was shown in overweight and obese subjects [106]. Moreover, in an epidemiological study on a cohort of overweight/obese Hispanic teenagers, a strong negative correlation between fructose consumption and *Firmicutes*, particularly *Eubacteria elegens*, was found [107]. In pregnant rats, high fructose intake modulated the maternal microbiota with a significant reduction in *Lactobacillus* and *Bacteroides* prevalence [90].

Moreover, in the same study, the impact of fructose intake was analyzed, not only in the pregnant rats but also in the offspring, showing in both of them a reduction of gene expression of epithelial intestinal tight junctions, suggesting that intestinal permeability is affected either by the fructose diet directly or through maternal gut related adaptations [90]. The gut microbiota catabolizes fructose, generating various metabolites, including short-chain fatty acids (SCFA), such as acetate, butyrate, and propionate, which have different functional roles in obesity, insulin sensitivity, and NAFLD pathophysiology [108]. The total amount of SCFA is higher in obese subjects in comparison to lean subjects [106]. In animal models of obesity and T2DM, oral and intravenous administration of butyrate and propionate may decrease the hepatic steatosis and improve the glucose tolerance, increasing the activity of liver AMPK phosphorylation and PPAR alfa genes involved in FFA oxidation and glycogen storage [108,109]. The fructose consumption influencing the gut microbiota may affect the composition of SCFA [96]. Precisely, acetate produced by fructose microbiota catabolism is converted in acetyl-CoA, feeding the hepatic lipogenesis (Figure 2) [110]. In fact, ATP citrate lyase knockout mice, upon high-fructose dietary regimens, develop NAFLD [110].

The depletion of microbiota using antibiotics treatments or the silencing of the hepatic acetyl-CoA synthetase short-chain family member 2 crucial for acetate catabolism suppress the conversion of fructose to acetyl-CoA and fatty acids [110]. The fructose finally may regulate the hepatic lipogenesis, promoting the expression of lipogenic genes but also providing carbon source via acetate. To support this notion, in rats with fructose-rich diets induced with metabolic syndrome, the use of antibiotics or fecal transplantations significantly reduced inflammation and oxidative stress [105]. Therefore, untargeted metabolomics-based studies are needed to discover novel microbiota metabolites associated with NAFLD incidence.

## 4. Future Perspectives: Developments in Treatment and Prevention of Fructose-Induced NAFLD

The cornerstone of NAFLD treatment is represented by lifestyle changes, including physical activity and balanced diet [111]. Twelve months of an intensive lifestyle intervention (ILI) on 5145 overweight or obese diabetic adults showed a decrease of steatosis (−50% vs. 22.8%) and incidence of NAFLD (3% vs. 26%) compared to patients who received only diabetes support and education (DSE), which is the standard behavioral intervention for patients with T2DM [112]. In NASH biopsy proven patients, it was shown that, after 48 weeks of lifestyle intervention using a combination of diet, exercise, and behavior modification, some markers of disease severity, such as NASH histological activity score, decreased [113]. Furthermore, it was proven in healthy subjects under high-fructose diets that exercise may prevent fructose-induced hypertriglyceridemia and promote a decrease of hepatic fat content independently from energy balance [114]. The dietary intervention for NAFLD prevention should be focused also on the reduction of fructose enriched beverages consumption. In fact, it was demonstrated that liquid sucrose diet-fed mice had higher hepatic triglycerides in comparison to their isocaloric solid sucrose-fed counterparts [115]. In a study focused on 359 children followed for 6 years, the development of insulin resistance signs, such as waist circumference/abdominal fat, was strongly linked to the intake of liquid sucrose compared to the total caloric intake [116].

Up to now, no pharmaceutical treatments are approved for NAFLD treatment, but a potential category of drugs can be effective in treating fructose-induced NAFLD [111]. One potential target of these therapeutic strategy includes KHK inhibitors [117]. As proof of the possible therapeutic efficacy of KHK inhibitors, it was demonstrated in mice that the KHK loss of function protects against the endogenous fructose production in which the conversion of serum glucose at high concentrations to fructose contributes to development of visceral obesity, fatty liver, and elevated insulin levels, even without excessive caloric intake [118].

In fact, fructokinase knockout mice (KHK-A and KHK-C) under high-fat/high-sugar diets developed obesity with mild hepatic steatosis, and there was no evidence of hepatic inflammation compared to wild-type mice [76]. In vitro, it was shown that pyrimidinopyrimidine compounds are selective and highly affine human KHK-C inhibitors, even if their fast clearance was also observed, which could limit the achievement of an effective blood therapeutic concentration in vivo [119]. In vitro, a novel series of indazoles proved KHK inhibition activities demonstrating ATP-binding capabilities, but studies on their therapeutic efficacy are still needed [120]. In a rat model discovery molecules study, a potential pyridine KHK inhibitor showed a favorable safety profile, resulting in a nonlinear decrease in F1P after fructose bolus [121]. On the basis of these promising in vitro and in vivo studies, KHK inhibitors may be applied in studies on humans in order to evaluate its potential therapeutic efficacy. Another therapeutic target is the inhibition of acetyl-CoA carboxylase (ACC) that modulates the conversion of acetyl-CoA to malonyl-CoA in order to limit the hepatic lipogenesis. In fact, in obese individuals with NAFLD, the hepatic de novo lipogenesis contributes up to 38% of intrahepatic triglycerides levels [122]. It was shown that a liver-specific inhibitor of ACC1 and ACC2 (MK-4074) administered for 1 month to subjects affected by hepatic steatosis lowered hepatic triglycerides levels and lipogenesis, inducing, however, hypertriglyceridemia and increasing VLDL secretion due to activation of lipogenic transcription factor SERBP-1c [123].

Other possible therapeutic alternatives are pro- and prebiotics due to their effects on GUT microbiota and/or intestinal barrier function. In a study on mice affected by fructose-induced NAFLD, it was shown that the addition of a probiotic bacterial strain of *Lactobacillus rhamnosus* GG (LGG) restored the duodenal tight junction protein expression, reducing LPS translocation and serum level of proinflammatory cytokines, such as TNF-alfa, IL-8, and IL-1beta [124].

Furthermore, the liver fat accumulation and the transaminase serum concentration were attenuated in fructose-induced NAFLD LGG fed mice [124].

An increasing interest is being posed on natural products and plant extracts (e.g., curcumin, resveratrol, and (-)-epicatechin) that could be effective in treatment of fructose-induced NAFLD [125]. In this field, it was shown that natural drugs such as *Symplocos cochinchinesis*, a popular Indian herbal medicine, and Silymarin, a flavonolignan isolated from *Silybum marianum,* decrease the expression of SERBP-1c and FAS, resulting in a downregulation of hepatic lipogenesis [126,127]. Curcumin, a phenolic compound isolated from *Curcuma longa*, in addition to effects oriented to reduce triglyceride hepatic content and decrease the expression of SREBP-1c, also showed a suppression activity of lipogenic enzymes, such as ACC and FAS, in rats treated with high-fructose diet [128]. Isorientin and (-)-epicatechin, flavonoids isolated from several edible plants, were shown to mitigate the fructose-induced metabolic disorders in high fructose-fed rats, enhancing antioxidant enzyme activities, inhibiting inflammation cytokines (TNF-alfa, IL-1, and IL-6), and ameliorating the liver injury [129,130]. Resveratrol, a phenol compound isolated from different edible plants, reduced the hepatic triglycerides content and improved insulin resistance in rats with hepatic steatosis induced by high-fructose corn syrup, not only suppressing SERBP-1C and FAS but also improving IRS, endothelial nitric oxide synthase, and sirtuin 1 [131]. However, the clinical effectiveness of these natural products remains to be evaluated.

Finally, insulin-sensitizing agents, such as metformin, have been proposed for the treatment of NAFLD. In mice fed with fructose solution, metformin showed a protective effect on the onset of fructose-induced NALFD and on the loss of thigh junction proteins occludin and zonula occludens in duodenum as well [132]. However, in NASH patients with DMT2, metformin did not show to improve histological alteration. Therefore, further evaluations are necessary in order to evaluate its therapeutic efficacy [111].

## 5. Conclusions

Fructose metabolism is implicated in the development and the progression of NAFLD, and the use of high fructose sweeteners, such as HFCS, is increasing in food industries due to its low cost and ease of addition to food products. The hepatic metabolism of fructose is unrestricted, bypassing the regulatory enzymes of glycolysis and resulting in potent induction of lipogenesis.

Moreover, the fructose, irrespective of negative feedback regulation of insulin signaling, induces indirectly an increase of hepatic de novo lipogenesis and gluconeogenesis via SREBP1c and ChREBP, contributing to an increase of hepatic insulin resistance. The fructose consumption contributes to hepatic inflammation and, consequently, NASH progression through the production of ROS, the accumulation of uric acid, and the induction of ER stress.

Furthermore, a high chronic fructose intake alters the GUT microbiota, resulting in microbial dysbiosis and altered intestinal permeability, which leads to bacterial and PAMPs translocation, inducing and promoting hepatic inflammation. Until now, animal studies have greatly improved our understanding of in vivo fructose metabolism and its casual effects on NAFLD, but in humans, several questions still remain open.

Therefore, clinical trials targeting fructose-induced NAFLD will be the key strategy to reveal and comprehensively understand the pathophysiology of the alterations induced by fructose and the underlying molecular mechanisms for clinical diagnosis and treatment.

## Figures and Tables

**Figure 1 nutrients-13-01314-f001:**
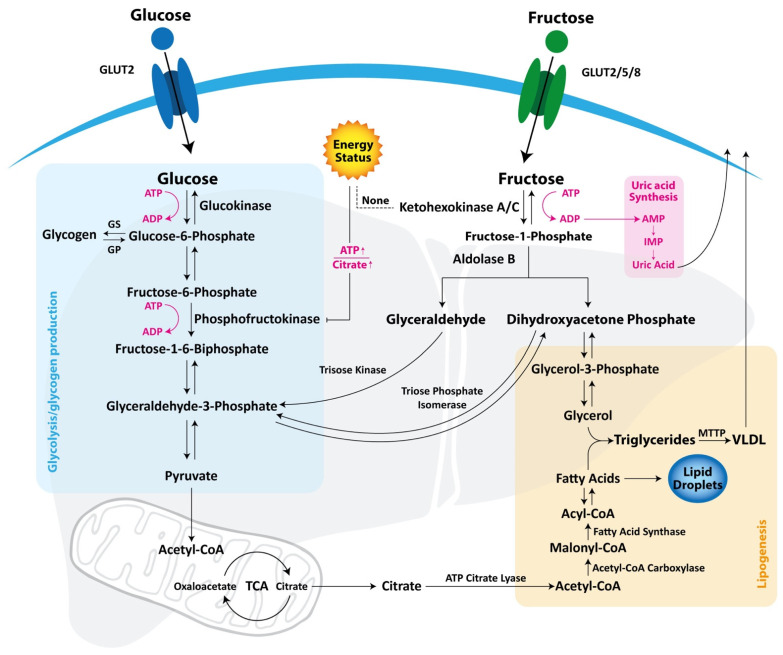
Glucose and fructose metabolism. Glucose is transported into the hepatocyte via GLUT 2 and is successively metabolized though glycolysis to produce ATP and pyruvate or is stored as glycogen. Fructose can be transported via GLUTs 2, 5, and 8 and is phosphorylated by ketohexokinase, A or C, into fructose-1-phosphate, that is cleaved in glyceraldehyde and dihydroxyacetone phosphate. While glucose metabolism is regulated by phosphofructokinase, which is inhibited by ATP and citrate when energy status is high, the ketohexokinase activity is unrestricted. Glyceraldehyde and dihydroxyacetone phosphate are phosphorylated, respectively, by triose kinase or triose phosphate isomerase in glyceraldehyde-3-phosphate to produce pyruvate for fuel triglyceride synthesis or fructose-1,6-biphosphate for gluconeogenesis. Dihydroxyacetone phosphate may be converted in glycerol-3-phosphate to production of triglycerides and VLDL lipoproteins. ADP, adenosine diphosphate; ATP, adenosine triphosphate, AMP, adenosine monophosphate; IMP, inosine monophosphate; Glut, glucose transporter; Gp, glycogen phosphorylase; Gs, glycogen synthase; MTTP, microsomal triglyceride transfer protein; TCA, citric acid cycle; VLDL, very low-density lipoprotein.

**Figure 2 nutrients-13-01314-f002:**
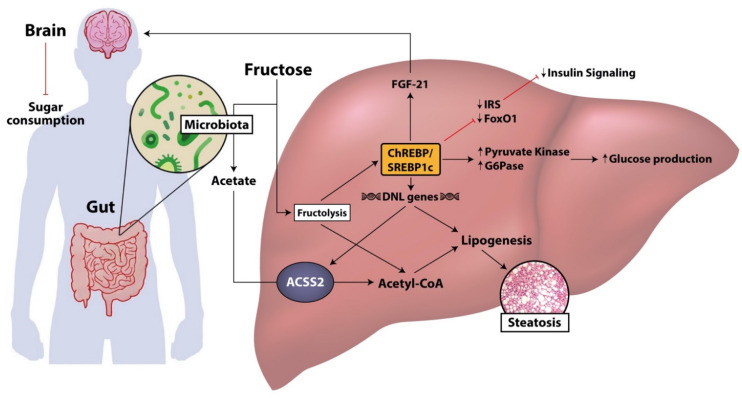
Fructose pathway involved in lipogenesis and insulin resistance. In the liver, fructose is metabolized as pyruvate via fructolysis, generating citrate and, successively, acetyl-CoA for lipogenesis. Dietary fructose is also converted by gut microbiota into acetate that is converted in acetyl-CoA by ACSS2 supplying the de novo lipogenesis. Fructose may directly activate SREBP1c and ChREBP, potent inducers of DNL genes such as ACSS2, contributing to the development of steatosis. ChREBP is active also on glucose production, inducing metabolic gene expression such as pyruvate kinase and glucose-6-phosphate, resulting in an increased gluconeogenesis. Furthermore, high fructose diet may reduce the expression of IRS and FoxO1, resulting in an inhibition of insulin signaling and an increased plasma glucose expression. On the contrary, as a compensatory mechanism, ChREBP activates the expression of several enzymes and hormones that modulate the sugar consumption, suppressing the sweet taste preference. ACSS2, Acetyl-CoA synthetase short-chain family member 2; ChREBP, carbohydrate-responsive element-binding protein; DNL, de novo lipogenesis; FGF-21, fibroblast growth factor—21; FoxO1, forkhead box O1; IRS, insulin receptor substrate; SREBP1c, sterol regulatory element-binding protein 1c.

**Figure 3 nutrients-13-01314-f003:**
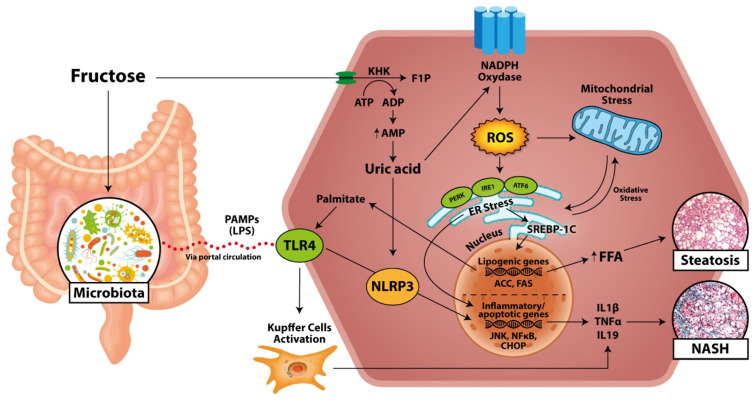
Fructose pathway involved in oxidative stress and inflammation. In the hepatocyte, fructose sugar is rapidly metabolized by KHK, depleting ATP and resulting in uric acid accumulation. Uric acid activates NADPH oxidase with production of ROS, which activates mitochondrial ROS production and stimulates UPR in ER. ER stress may also activate mitochondrial ROS production and vice versa. Moreover, ER stress contributes to progression of hepatic steatosis activating lipogenic genes via PERK, ATF4, and IRE1 and inducing SREBP-1c that increases the activity of ACC and FAS. As a response of excessive oxidative stress, inflammatory and apoptotic pathways are activated through JNK, NFkB, and CHOP, resulting in an increased production of cytokines (IL1ß, TNFa, IL19), which plays an important role in the development of NASH. Fructose consumption may alter GUT microbiota and intestinal permeability, increasing endotoxin translocation of, for example, LPS, the most common PAMPs, into the liver via portal circulation. PAMPs may activate TLR4 that promotes the Kupffer activation and, subsequently, inflammatory pathways. Altered lipid homeostasis and fatty acids such as palmitate, in which production is promoted by fructose, may activate TLR4. TLR4, but also the excess of fat, activate the inflammasome involved in production of inflammatory cytokines. ACC, Acetyl-CoA carboxylase; ATF4, activating transcription factor 4; CHOP, CCAAT/enhancer-binding homologous protein; IRE1, inositol-requiring signaling protein 1; IL, interleukin; FAS, fatty acids synthase; JNK, c-Jun N-terminal kinase; KHK, ketohexokinase; FFA, free, fatty acids; LPS, lipopolysaccharide; NASH, non-alcoholic steatohepatitis; NFkB, nuclear factor kB; NLRP3, PAMPs, pathogen-associated molecular patterns; PERK, protein kinase RNA-like ER kinase; SREBP1c, sterol regulatory element-binding protein 1c; TNFα, tumor necrosis factor alfa; TLR4, toll-like receptor 4.

## Data Availability

Not applicable.

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
