# Peer review of "The Role of Fructose in Non-Alcoholic Steatohepatitis: Old Relationship and New Insights"

_nutrients, 2021, doi:10.3390/nu13041314_

Round 1

Reviewer 1 Report

This Review discusses the effects of Fructose in NAFLD/NASH development starting from biochemical mechanisms (Fig. 1) to the role of microbiota to de novo lipogenesis (Fig.2) and oxidative stress and Inflammation (Fig.3).

  1. In Fig. 1, the characters are quite faint and light, it is difficult to read. perhaps, increase font size and bolded Fonts.
  2. In Fig. 2 legend, mistake in 'expressScheme21'.
  3. In Fig. 3 legend,mistake 'Scheme3' 
  4. Line 238, 'is concerend' not concerns.
  5. Line 240, decreasing not decrease.
  6. 'has been shown' not has been showed in line 430, 439, 523, 536, 542, 550, 555, 567, 570.
  7. Line 499, NAFLD incidence
  8. of of in Line 507
  9. ad in line 579

Author Response

we thank the Reviewer for her/his comments. They resulted useful to improve the quality of the paper.

  1. As suggested, we modified the font size in order to improve the readability of the figure.

2-9. We corrected the highlighted mistakes

Reviewer 2 Report

In this manuscript entitled " The role of fructose in non-alcoholic steatohepatitis: old relationship and new insights ", the authors summarized some risk factors of fructose on etiology of NAFLD/NASH.

Minor points:

The work presented has a lot of typographical errors.

  1. Figure1 pyuvate: pyruvate
  2. Figure2 Insuline: Insulin
  3. Figure2 Piruvate: Pyruvate
  4. L216 expresScheme 21: I don't understand the meaning of this word
  5. L219 1c: 1c
  6. L304 excesScheme 3.: I don't understand the meaning of this word

Author Response

we thank the Reviewer for her/his comments. They resulted useful to improve the quality of the paper. We corrected all the highlighted mistakes both in the figures and main text.